# Economic Evaluation of Hepatitis C Treatment Extension to Acute Infection and Early-Stage Fibrosis Among Patients Who Inject Drugs in Developing Countries: A Case of China

**DOI:** 10.3390/ijerph17030800

**Published:** 2020-01-28

**Authors:** Yin Liu, Hui Zhang, Lei Zhang, Xia Zou, Li Ling

**Affiliations:** 1Department of Medical Statistics, School of Public Health, Sun Yat-Sen University, Guangzhou 510080, China; liuy429@mail2.sysu.edu.cn (Y.L.); zouxia@mail3.sysu.edu.cn (X.Z.); 2Sun Yat-sen Center for Migrant Health Policy, Sun Yat-Sen University, Guangzhou 510080, China; 3Department of Health Policy and Management, School of Public Health, Sun Yat-Sen University, Guangzhou 510080, China; zhanghui3@mail.sysu.edu.cn; 4China-Australia Joint Research Center for Infectious Diseases, School of Public Health, Xi’an Jiaotong University Health Science Center, Xi’an 710000, China; lei.zhang1@xjtu.edu.cn; 5Melbourne Sexual Health Center, Alfred Health, Melbourne VIC 3053, Australia; 6Central Clinical School, Faculty of Medicine, Monash University, Melbourne, VIC 3800, Australia; 7Department of Epidemiology and Biostatistics, College of Public Health, Zhengzhou University, Zhengzhou 450001, China

**Keywords:** hepatitis C virus, cost-effectiveness, antiviral treatment, people who inject drugs

## Abstract

We aimed to assess the cost-effectiveness of (1) treating acute hepatitis C virus (HCV) vs. deferring treatment until the chronic phase and (2) treating all chronic patients vs. only those with advanced fibrosis; among Chinese genotype 1b treatment-naïve patients who injected drugs (PWID), using a combination Daclatasvir (DCV) plus Asunaprevir (ASV) regimen and a Peg-interferon (PegIFN)-based regimen, respectively. A decision-analytical model including the risk of HCV reinfection simulated lifetime costs and quality-adjusted life-years (QALYs) of three treatment timings, under the DCV+ASV and PegIFN regimen, respectively: Treating acute infection (“Treat at acute”), treating chronic patients of all fibrosis stages (“Treat at F0 (no fibrosis)”), treating only advanced-stage fibrosis patients (“Treat at F3 (numerous septa without cirrhosis)”). Incremental cost-effectiveness ratios (ICERs) were used to compare scenarios. “Treat at acute” compared with “Treat at F0” was cost-saving (cost: DCV+ASV regimen—US$14,486.975 vs. US$16,224.250; PegIFN-based regimen—US$19,734.794 vs. US$22,101.584) and more effective (QALY: DCV+ASV regimen—14.573 vs. 14.566; PegIFN-based regimen—14.148 vs. 14.116). Compared with “Treat at F3”; “Treat at F0” exhibited an ICER of US$3780.20/QALY and US$15,145.98/QALY under the DCV+ASV regimen and PegIFN-based regimen; respectively. Treatment of acute HCV infection was highly cost-effective and cost-saving compared with deferring treatment to the chronic stage; for both DCV+ASV and PegIFN-based regimens. Early treatment for chronic patients with DCV+ASV regimen was highly cost-effective.

## 1. Introduction

Hepatitis C virus (HCV) infection remains the leading cause of liver cirrhosis or hepatocellular carcinoma [1], and has a substantial negative impact on patients’ quality of life and functioning [2,3,4]. HCV infection can be classified as acute or chronic infection. The acute infection was defined as a new infection occurring from the time of exposure up through six months of infection [5]. Approximately 26% (95% confidence interval [CI]: 22–29%) of the acute-infected individuals would spontaneously clear infection, otherwise the chronic infection would subsequently develop [5,6]. 

People who inject drugs (PWID) were one of the most at-risk populations for HCV [1]. In 2017, there were an estimated 15.6 million PWID aged 15–64 years worldwide, and among them 52% (95% CI: 42–62%) were infected with HCV [7], accounting for 8% of current HCV infections globally [1]. PWID also accounted for the biggest proportion (estimated 23%) of acute HCV infections worldwide [1], and were at an increased risk of reinfection from HCV and excess mortality from drug overdose [8,9].

To eliminate HCV, the World Health Organization (WHO) put forward a 2030 target that 80% of diagnosed HCV cases would be treated, and suggested that PWID should be prioritized for treatment [10]. China has the largest number of PWID in the world, with an estimated HCV seroprevalence of 67.0% [11], and approximately 25% new infection [12,13]. Oral direct-acting antivirals (DAAs) regimen and the Peg-interferon (IFN)-based regimen are two types of antiviral drugs for the treatment of HCV. Due to the expensive nature of DAAs, in China and many other resource-limited settings where DAAs are newly or not yet approved, PegIFN-based regimen is still the frontline HCV therapy [14]. PegIFN as a monotherapy is the frontline therapy for acute HCV infection, while combining PegIFN with ribavirin (RBV) is the frontline therapy for chronic HCV infection [14]. In April 2017, the combination of Daclatasvir (DCV)+Asunaprevir (ASV) as the first generic DAAs was approved in China, and it represented a treatment option for chronic HCV genotype 1b infection [15], the predominant genotype (56.8%) in China [16].

Due to economic issues and limited health care infrastructure in China, as in other resource-limited settings, policy makers should determine which subgroup should be prioritized for HCV treatment. Regarding acute HCV infection among PWID [11], PegIFN and DAAs therapies have already been demonstrated to be with higher efficacy and shorter-duration than treating chronic HCV infection [17]. However, current clinic practice recommends deferring treatment to the chronic stage or monitoring patients with acute HCV for at least 12–24 weeks [5]. On one hand, treating acute HCV infection may lead to unnecessary additional costs and over-treatment, because some individuals would spontaneously clear infection and have no symptoms [6]. On the other hand, the cost-effectiveness of early treatment of acute infection was not confirmed. Only one modeling-based study from the United States suggested that treatment of acute HCV infection under the DAAs regimen was cost-effective compared with deferring treatment to the chronic stage, but it did not include HCV reinfection in the model [5]. This may result in underestimating or overestimating the benefits of treatment for acute infection among PWID patients. What’s more, the results might not necessarily be applied to the developing countries as the cost-effectiveness of HCV treatment was influenced by the cost of treatment and the risk of reinfection [18]. Therefore, it is necessary to confirm the cost-effectiveness of initiating therapy in the acute vs. deferring until the chronic phase of HCV infection, among PWID patients, especially in developing countries.

Regarding chronic HCV infection among PWID, the WHO recommends “treat all” based on previous cost-effectiveness studies, which suggested initiating treatment for chronic HCV regardless of the fibrosis stages [1]. However, these cost-effectiveness analyses were all conducted in developed countries, such as Australia [19,20,21], the Netherlands [22], the United Kingdom [23,24,25] and Italy [26]. Evidence on the cost-effectiveness of expanding treatment to the early fibrosis stage related to access to HCV treatment was substantially more limited in developing countries. Only one study in Egypt [27] suggested that immediate treatment of patients with early fibrosis stage was less expensive and more effective than delaying treatment, but it did not include the possibility of HCV reinfection in the model. This omission may result in underestimating or overestimating the benefits of early treatment for chronic infection [1]. In China, the oral DAAs have been confirmed to be cost-saving and cost-effective compared with the PegIFN-based regimen [28], but the cost-effectiveness of initiating treatment at an early fibrosis stage compared with advanced fibrosis stage was not reported. Furthermore, previous studies in developed and developing countries focused only on either DAAs or PegIFN-based therapy for HCV. They did not indicate how to adjust treatment strategy to local conditions in the presence of both drugs.

Thus, this study used the case of China to conduct a model-based analysis to examine the cost-effectiveness of different treatment timings under DCV+ASV and PegIFN-based regimens, respectively. With a hypothesis that early treatment was more effective, we first compared treatment at acute with deferring treatment until stage F0, to determine the cost-effectiveness of treating acute HCV, under DCV+ASV and PegIFN-based regimens, respectively. Then, we compared treatment at stage F0 with until stage F3, to assess the cost-effectiveness of early treatment for chronic HCV among PWID, under DCV+ASV and PegIFN-based regimens, respectively.

## 2. Materials and Methods

### 2.1. Model Overview

Based on natural history of HCV infection, we considered regression of liver damage and HCV reinfection among PWID. We developed a Markov cohort, state-transition model to evaluate the health outcomes and costs of treatment at different timing with DCV+ASV or PegIFN-based regimen. This model has been received by decision makers and clinicians [5,29]. The disease stage-transition reflect progression through the acute phase or 5 METAVIR (Meta-analysis of Histological Data in Viral Hepatitis) liver fibrosis stages (F0, no fibrosis; F1, portal fibrosis with septa; F2, portal fibrosis with rare septa; F3, numerous septa without cirrhosis; F4, compensated cirrhosis) to advanced liver disease (DC, decompensated cirrhosis; HCC, hepatocellular carcinoma; LT, liver transplantation; and Post-LT) and regression of liver damage or reinfection after patients cleared the virus. We simulated the clinical course of the patients and projected long-term outcomes such as quality-adjusted life years (QALYs) and costs. The model ran in a monthly cycle length until all patients died. Additional details were provided in Figure 1, and input parameters were summarized in Appendix A.

### 2.2. Natural History of HCV Infection

Acute HCV patient who failed to spontaneously clear the virus or be cured would progress to F0 fibrosis stage after six months. Patients at F0 who failed to be cured progressed through different stages of liver fibrosis (F0–F4). Patients with F4 could further progress to DC, HCC. Transition probabilities between states were obtained from published systematic reviews and observational studies. Patients with DC and HCC were eligible for liver transplantation. The likelihood of LT was estimated from previous studies as 0.008 (95% CI: 0.006–0.01) (Appendix A).

### 2.3. Patient Cohort

Our base-case cohort was representative of newly diagnosed and treatment-naïve genotype 1b HCV RNA positive PWID in China. Patients coinfected with HIV or HBV were excluded. The mean age of the cohort at the baseline was 20.7 years old [30]. Distribution of HCV stages at baseline was calculated on the basis of previous investigation in China [12,13]: Acute 25%, F0 22.5%, F1 17.2%, F2 7.5%, F3 9%, and F4 18.8%. The model did not distinguish patients from viral concentration, sex, or race, although these factors may affect treatment outcomes [29].

### 2.4. Progression, Regression and Reinfection after SVR

Patients who cleared virus entered the recovered states, and some patients could experience regression of liver fibrosis. Despite being cured, patients in the F4_SVR state risked progression to DC or HCC, but with a greatly reduced rate [31,32]. To make a conservative estimation of the cost-effectiveness of treatment in PWID, we stated that reinfection would occur at a high proportion of 19.0% (range: 0.6–19.0%) [8,33]. Those re-infected patients would spontaneously re-clear the virus with a proportion of 52% (95% CI: 33–73%) [34], then other patients would re-enter the Markov HCV progression to the chronic HCV state to receive treatment. Those who failed treatment were not eligible for re-treatment. Previous studies had suggested that broadly expanded treatment could provide substantial health gains due to the reduced the reinfection risks [18]. In this study, due to the lack of data in China, we made a relatively conservative assumption of the expanding treatment’s potential for reducing HCV reinfection as 5% (range: 2.5–7.5%) [5], one-way sensitivity analysis would be conducted to determine its impact on cost-effectiveness.

### 2.5. Mortality

Mortality rates were shown in Appendix A. Besides the age-specific background mortality for the general population from China 2017 Life Tables [35], overdose of drug-related mortality from a systematic review and meta-analysis of cohort studies was also included [9]. Mortality for patients with acute HCV, stage F0 to F4, and clearing the virus was assumed to be the background mortality rate plus overdose of drug related mortality rate. Patients with DC and HCC had excessive liver-related mortality [36,37]. Patients who receive a liver transplant and who were after transplantation could also die from transplant-related complications [38].

### 2.6. Treatment Strategies

Three options of initiating treatment at different timing under two therapy options (DCV+ASV and PegIFN-based regimen) were modeled (Figure 1A).

(i) Treat at acute. In this scenario, all acute infections could be immediately treated. With DCV+ASV, the course was 12 weeks, referred to as “DCV+ASV (treat at acute)” arm. With PegIFN-α as monotherapy, the course was 24 weeks, referred to as “PegIFN (treat at acute)” arm.

Due to the lack of data on DCV+ASV in the treatment of acute HCV infection, considering acutely-infected patients have historically been treated with a shorter duration, we conservatively assumed that the duration of treating acute infections using DCV+ASV was 12 weeks based on a published review [17].

(ii) Treat at F0. In this scenario, the acute-infected patients must progress to stage F0 to be treated. All patients who had developed chronic HCV infection could be treated regardless of the stage of fibrosis (F0–F4). With DCV+ASV, the course was 24 weeks, referred to as “DCV+ASV (treat at F0)” arm. With PegIFN+RBV, the course was 48 weeks, referred to as “PegIFN+RBV (treat at F0)” arm.

(iii) Treat at F3. In this scenario, only those patients at F3 and F4 stages could be treated. Those at acute phase or stage F0–F2 had to progress to stage F3 to be treated. All patients at F3 and F4 stages could be treated. With DCV+ASV, the course was 24 weeks, referred to as “DCV+ASV (treat at F3)” arm. With PegIFN+RBV, the course was 48 weeks, referred to as “PegIFN+RBV (treat at F3)” arm.

The goal of treatment was an undetectable serum level of HCV RNA 12 or 24 weeks after the completion of therapy, named a sustained virologic response (SVR). Due to data unavailability, we conservatively estimated that the SVR rate in acute treatment was equivalent to that in chronic treatment using the DCV+ASV regimen. The SVR of perIFN monotherapy was between 71% and 94% in patients with acute HCV mono-infection based on a published review [17]. Discontinuation of therapy was considered given the poor adherence to treatment among PWID [39,40,41,42].

### 2.7. Costs and Health State Utility Values

The societal perspective of Chinese was adopted to calculate all direct medical costs for HCV management and therapy (Appendix A). All outcomes were presented on per cohort basis.

All costs were expressed as US dollars using official exchange rates as of 2018 (US$1 = 6.62 CNY) [43]. Regimen costs of pegIFN-based and DCV+ASV regimens were derived from previous economic evaluations in a Chinese setting [42,44]. The annual direct medical costs of managing patients at stages F0–F4, DC, and HCC were obtained from a real-world study in China [44], which included the costs of outpatient visits and post-treatment monitoring. The annual costs of liver transplantation and post-liver transplantation were gained from literature focused on health costs among chronic hepatitis B infection in China [45]. HCV-RNA and genotype tests were used to confirm infection. HCV-RNA test was also used to determine whether spontaneous clearance had occurred, and the relevant costs were obtained from local charges.

The model included health state utility values by fibrosis stages with and without SVR and disutility during treatment. Utility values were obtained from previous literatures based on SF-36 [28,46,47,48]. We assumed the utility value for the acute phase of HCV was equal to that for the F0 states, and post SVR or spontaneously cleared virus patients from acute phase was equal to that for the F0_SVR patients.

### 2.8. Model Outcomes and Statistical Analysis

Statistical analysis and Markov model were performed using TreeAge Pro 2018, and graph-plotting was done with Excel software. All future costs and QALYs were discounted at 5% (3–5%) per year.

Incremental cost-effectiveness ratios (ICERs) as the ratio of the difference in costs between treatment strategies divided by the difference in QALYs were calculated. A strategy producing an ICER of US$29,295 per QALY, as 3-times per capita gross domestic product (GDP) of China in 2018 [43], was considered as cost-effective. A strategy producing an ICER of US$9765 per QALY, as one-time per capita GDP of China in 2018, was considered as being highly cost-effective.

One-way sensitivity analysis was conducted to determine the effects of parameters on the ICER. Probabilistic sensitivity analysis based on a second-order Monte Carlo simulation with 1000 iterations was then conducted to ascertain the model stability. Results were reported as cost-effectiveness acceptability curves. The range, distribution, and source for each parameter were shown in Appendix A.

## 3. Results

### 3.1. Base-Case Results

#### 3.1.1. Cost-Effectiveness of Treating Acute HCV among PWID

Treating acute infection was cost-saving and more effective compared with delayment to the F0 stage (Table 1).

With the DCV+ASV regimen, treatment at acute gained an extra QALY of 0.007 and reduced the cost by US$1737.275. With PegIFN regimen, treating at acute gained an extra QALY of 0.032 and reduced the cost by US$2366.790.

#### 3.1.2. Cost-Effectiveness of Early Treatment at F0 Stage for Chronic HCV among PWID

Early treatment at stage F0 cost more and resulted in a gain of QALY (Table 2).

With DCV+ASV regimen, early treatment at F0 compared with waiting until stage F3 increased QALY by 0.459 and costs by US$1735.488, the corresponding ICER was US$3780.20/QALY, which was below 1-time per capita GDP (US$9765/QALY). This indicated that early treatment with DCV+ASV for chronic HCV was highly cost-effective.

With PegIFN+RBV regimen, early treatment at F0 compared with delayed treatment at F3 had a QALY gain of 0.345 but with a higher cost of US$5225.364, the corresponding ICER was US$15,145.98/QALY, which was below 3-times per capita GDP (US$29,295/QALY) but above 1-time per capita GDP. This indicated that early treatment with PegIFN+RBV for chronic HCV was cost-effective, but the cost-effectiveness was not high.

### 3.2. One-Way Sensitivity Analysis

#### 3.2.1. Cost-Effectiveness of Treating Acute HCV among PWID

Treating acute infection compared with deferring treatment until stage F0 had substantially lower costs and more QALYs across all parameters’ ranges, regardless of the drug regime—DCV+ASV or PegIFN. The ICER was most sensitive to the reinfection rate after clearing the virus. With the reduction of reinfection rate, treating acute infection cost less and gained more QALYs (Appendix A).

#### 3.2.2. Cost-Effectiveness of Early Treatment at F0 Stage for Chronic HCV among PWID

With the DCV+ASV regimen, the ICER of treatment at stage F0 vs. at stage F3 was most sensitive to the reinfection reduction rate from treatment, the costs of DCV+ASV, the reinfection rate after clearing virus, and the utility of F2. While the reduction probability was set at 5%, the ICER was above one-time per capita GDP but not higher than three-times per capita GDP. With the PegIFN+RBV regimen, the ICER of treatment at F0 vs. at F3 was most sensitive to the SVR of PegIFN+RBV, the utility of F2, the reinfection reduction rate from treatment, the costs of PegIFN+RBV, the utility of F0_SVR, the reinfection rate after clearing virus and the re-clearance proportion within six months after reinfection. The ICER was less than one-time per capita GDP, while the reinfection rate was lower than about 12.8%. The ICER was above three-times per capita GDP when SVR of PegIFN+RBV was below about 0.4, utility of F2 was above 0.993, and the utility of F0_SVR was below approximately 0.981 (Appendix A).

### 3.3. Probabilistic Sensitivity Analysis

Probabilistic sensitivity analysis demonstrated that the base-case analysis was stable.

Monte-Carlo simulations were shown in Figure 2 and Figure 3 and Appendix A as the likelihood of a strategy to be considered cost-effective at different willingness-to-pay (WTP). Treatment at the acute stage was cost-effective in 100% of simulations compared with deferring treatment until stage F0, no matter whether it was with the DCV+ASV or PegIFN-based regimen (Appendix A). For chronic HCV, at a WTP threshold of one-time per capita GDP, early treatment at stage F0 compared with delayed treatment at stage F3 was cost-effective in 100% of simulations under DCV+ASV regimen, but delayed treatment at stage F3 was cost-effective in 54.6% of simulations under PegIFN+RBV regimen; at a WTP threshold of three-times per capita GDP, early treatment at stage F0 compared with delayed treatment at stage F3 was cost-effective in 87.4% of simulations under PegIFN+RBV regimen (Appendix A).

Compared to the other strategies, treatment at acute using DCV+ASV was the cost-effective strategy, with a high probability of 100% at a WTP threshold of one-time per capita GDP (Figure 3). When only PegIFN-based regimen was available, treatment at acute was also the cost-effective option compared to the other strategies, with a probability of 88.8% at the threshold of one-time per capita GDP and 98.9% at the threshold of three-times per capita GDP (Figure 3).

## 4. Discussion

This study used a Markov model to assess the cost-effectiveness of treating acute HCV and early treatment for chronic HCV, under DCV+ASV and PegIFN-based regimen, respectively. This study demonstrated that treatment at an acute stage compared with deferring until chronic stage was highly cost-effective or cost-saving, for both DCV+ASV and PegIFN-based regimens. With the threshold of one-time per capita GDP of China, early treatment at F0 stage was cost-effective compared with delayed treatment at F3 stage using DCV+ASV but not cost-effective when using the PegIFN-based regimen. When the threshold was set at three-times per capita GDP of China, early treatment at F0 stage was cost-effective for both the DCV+ASV and PegIFN-based regimens. This provided new evidences for improving current treatment guidelines that suggested deferring treatment to the chronic stage for an acute infection. This was the first study which included the risk of HCV reinfection in the model to assess the cost-effectiveness of early treatment vs. delayed treatment among chronic HCV in developing countries.

Treatment at the acute stage was the most cost-effective option, especially using DCV+ASV. In line with a study conducted in the United States [5], treatment at an acute stage was highly cost-effective and cost-saving regardless of the reinfection rate or the costs of treatment. Actually, estimates for the efficiency and costs of acute treatment in this study were likely to be conservative, and treating acute HCV in developing countries may in fact be more cost-effective than we predicted. Thus, acute-infected individuals should not be deprived of being treated since spontaneous clearance of the virus may occur in some patients. It may be time to revise treatment guidelines to recommend treating acute HCV rather than deferring treatment to the chronic stage, regardless of the population (such as PWID, general population), the regimen (DAAs or PegIFN-based), and the setting (resource-limited settings and resource-abundant settings). However, early diagnosis was hard to reach, especially for PWID, because of discrimination, criminalization, and stigma associated with abusing drugs [49]. Optimizing the impact of effective treatment might require more interventions to facilitate access to early HCV detection, including promoting health awareness, addressing discrimination and stigma, regular testing for HCV, and so on [50,51].

Consistent with previous cost-effectiveness studies of treating chronic HCV among PWID in developed countries [18,19,23,26], we found that early treatment at F0 was slightly effective and more expensive than delayed treatment at the F3 stage. Whether early treatment was cost-effective or not compared with delayed treatment depended on the national per capita GDP. For example, in Australia, early treatment at F0 was considered cost-effective compared with delayed treatment at F3 with the threshold of AUD$50,000 per QALY, no matter using DAAs [19] or PegIFN+RBV regimen [20]. In this study, with the threshold of one-time per capita GDP of China (US$9765 per QALY), early treatment at F0 compared with delayed treatment at F3 stage was considered as cost-effective using DCV+ASV but not cost-effective when using the PegIFN-based regimen. This result could provide evidence for China and other resource-limited countries to optimize the allocation of medical resources. In those resource-limited settings, especially for low- and middle-income countries, when DAAs were not available, prioritized treatment for chronic patients with advanced-stage fibrosis may be a better option.

Moreover, this study confirmed that the cost-effectiveness of early treatment for chronic patients was sensitive to “reinfection rate after clearing virus” and “reinfection reduction rate from treatment”, which were two input variables for the model. The lower the reinfection rate, the more cost-effective “treat at F0” was. The reduction rate reflected the potential for reducing the risk of reinfection and secondary transmission from other infected individuals [18]. We also found that with the improvement of the potential, “treat at F0” was more cost-effective. This also suggested that expanding access to HCV treatment should be combined with harm reduction programs such as needle exchange and opiate substitute treatment, since they would complement HCV treatment by reducing reinfection risk for PWID [18]. The cost-effectiveness of early treatment in chronic patients was also subjected to treatment costs, but early treatment was still highly cost-effective even when the weekly costs of DCV+ASV was US$431.4. Actually, the price of highly effective DAAs has decreased substantially [18]. Early treatment using DAAs might be more cost-effective in the future.

This study has some limitations. First, some of the model inputs were obtained from literatures published worldwide, which may not reflect China-specific data. Second, our model focused on the overall simulation on population-level natural history, thus individual heterogeneity was only represented by varying some parameters in sensitivity analyses. Third, the model did not consider those patients with repeated treatment due to poor response. Fourth, we did not consider other genotype patients, and other DAAs approved in China, such as ombitasvir/paritaprevir/ritonavir+dasabuvir for genotype 1b, and sofosbuvir+velpatasvir/daclatasvir for all genotypes [28]. The efficiency and costs of these regimens were similar to DCV+ASV, but their treatment course was shorter. Therefore, the cost-effectiveness of other DAAs regimens for treating HCV in PWID would be consistent with DCV+ASV. Fifth, indirect medical costs were not considered, which may overestimate the cost-effectiveness of HCV treatment in PWID. Finally, our model only considered the patients mono-infected with HCV, excluding those coinfected HBV or HIV. Despite these limitations, we believe the conclusion would not be changed regardless.

## 5. Conclusions

In conclusion, treatment of acute HCV infection was highly cost-effective and cost-saving compared with deferring treatment to the chronic stage, for both DCV+ASV and PegIFN-based regimens. For patients who have been chronically infected, early treatment for chronic patients with DCV+ASV regimen was highly cost-effective. In those resource-limited settings where DCV+ASV or other DAAs were not available, prioritized treatment for those with advanced-stage fibrosis may be a better option. In the future, some real-world studies are needed to confirm and quantify the effects of HCV treatment in mathematical modeling studies. Similarly, it is also important to further examine the cost-effective of HCV treatment in other developing countries, especially in low- and middle-income countries.

## Figures and Tables

**Figure 1 ijerph-17-00800-f001:**
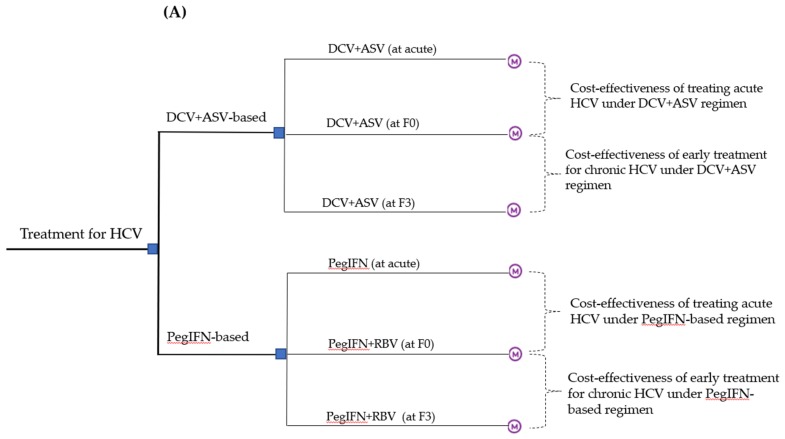
Decision analytical Markov model. (**A**) depicts treatment strategies with consideration for therapy and timing, each strategy analysis starts at the node marked with an ‘M’; (**B**) shows the Markov model. The Markov model structure is similar for all strategies, within each strategy, the fibrosis state at which the treatment that was initiated was selected. The square in the Markov model represents initial cohorts. HCV, Hepatitis C Virus; DCV, Daclatasvir; ASV, Asunaprevir; IFN, interferon; RBV, ribavirin; SVR, sustained virologic response; F0, no fibrosis; F1, portal fibrosis with septa; F2, portal fibrosis with rare septa; F3, numerous septa without cirrhosis; F4, compensated cirrhosis; DC, decompensated cirrhosis; HCC, hepatocellular carcinoma; LT, liver transplantation.

**Figure 2 ijerph-17-00800-f002:**
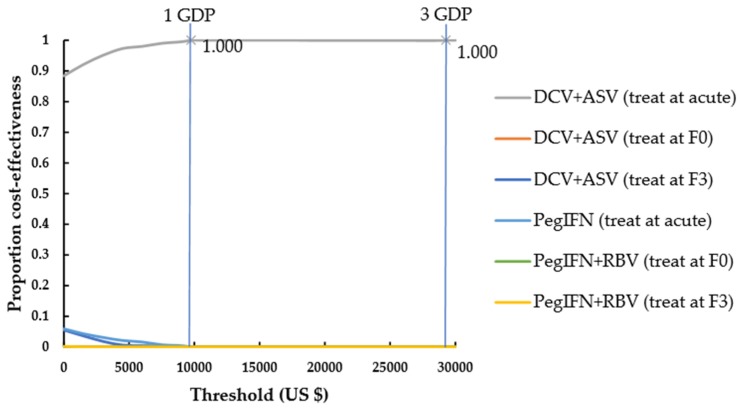
Cost-effectiveness acceptability curves for all strategies. F0, no fibrosis; F3, numerous septa without cirrhosis; GDP, gross domestic product. IFN, interferon; DCV, Daclatasvir; ASV, Asunaprevir; RBV, ribavirin.

**Figure 3 ijerph-17-00800-f003:**
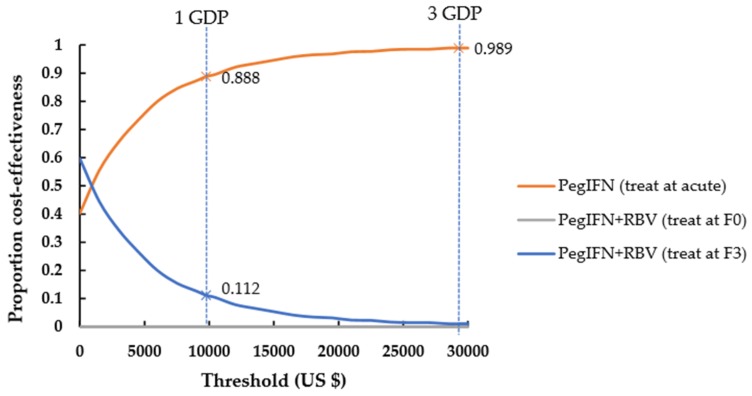
Cost-effectiveness acceptability curves when only PegIFN-based regimen was available. F0, no fibrosis; F3, numerous septa without cirrhosis; GDP, gross domestic product. IFN, interferon; DCV, Daclatasvir; ASV, Asunaprevir; RBV, ribavirin.

**Table 1 ijerph-17-00800-t001:** Costs, QALY, and ICER of treating acute HCV compared with delayed to F0 stage under DCV+ASV and PegIFN-based regimen.

Treatment Strategy	Total Treatment Costs/Person (US$)	QALYs/Person	Incremental Costs/Person (US$)	Incremental QALYs/Person	ICER (Incremental Costs/Incremental QALYs)
**DCV+ASV-based regimen**					
DCV+ASV (at F0)	$16,224.250	14.566	-	-	-
DCV+ASV (at acute)	$14,486.975	14.573	−$1737.275	0.007	dominant
**PegIFN-based regimen**					
PegIFN+RBV (at F0)	$22,101.584	14.116	-	-	-
PegIFN (at acute)	$19,734.794	14.148	−2366.790	0.032	dominant

Note: 5% discount rate applied to both costs and outcomes. QALY, quality-adjusted life year. ICER, incremental cost-effectiveness ratios.

**Table 2 ijerph-17-00800-t002:** Costs, QALY, and ICER of Early treatment at stage F0 compared with delayed to F3 stage under DCV+ASV and PegIFN-based regimen.

Treatment Strategy	Total Treatment Costs/Person (US$)	QALYs/Person	Incremental Costs/Person (US$)	Incremental QALYs/Person	ICER (Incremental Costs/Incremental QALYs)
**DCV+ASV-based regimen**					
DCV+ASV (at F3)	$14,488.762	14.107	-		-
DCV+ASV (at F0)	$16,224.250	14.566	$1735.488	0.459	$3780.20
**PegIFN-based regimen**					
PegIFN+RBV (at F3)	$16,876.220	13.771	-	-	-
PegIFN+RBV (at F0)	$22,101.584	14.116	$5225.364	0.345	$15,145.98

Note: 5% discount rate applied to both costs and outcomes. QALY, quality-adjusted life year. ICER, incremental cost-effectiveness ratios.

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
