# Peer review of "Economic Evaluation of Hepatitis C Treatment Extension to Acute Infection and Early-Stage Fibrosis Among Patients Who Inject Drugs in Developing Countries: A Case of China"

_ijerph, 2020, doi:10.3390/ijerph17030800_

Round 1

Reviewer 1 Report

I think this modeling was well done, but results are presented incorrectly. ICERs should only be calculated for non-dominated strategies, and these should be shown in the results table (not the current unclear CER). Also, this model does not include dynamic transmission, which is misleading based on the abstract and introduction.

Abstract:

The framing of the interventions is confusing. It would be more clear to describe the 3 treatment timings and 2 drug regimens, then frame as 6 comparator interventions.

Introduction:

Paragraph 1: Add how many people spontaneously clear the acute infection.

Line 57: Clarify 25%. Does this mean 25% of prevalent infections are acute?

Line 63: Is this approved for acute, chronic, or both?

Line 68-69: Higher efficacy and shorter duration compared to what?

Line 74: Do you mean early treatment may be cost-effective? Saying limited studies sounds like you are hypothesizing it will not be cost-effective.

Line 99-103: The framing of the comparators in unclear. Is this two separate comparisons, meaning you are not comparing (1) to (2)? Can all each drug regimen be used in each treatment strategy? It would be more clear to frame as 2 interventions strategies and compare them all.

Methods

Section 2.1: Are you including all genotypes in the population, or just 1b?

Section 2.3: Are you assuming 100% coverage, meaning for each treatment strategy everyone in the relevant disease stage gets treated?

Section 2.4: What to probability of getting a transplant?

Section 2.5: The reinfection is not clear. In Figure 1, it looks like reinfection would cause the person to go back to the same stage as they were before treatment. For spontaneously cleared, it looks like reinfection goes to F0. Shouldn’t all reinfection start as acute, and have some probability of clearing spontaneously?

Section 2.7: Societal perspective or healthcare perspective?

Are outcomes presented on a per person or per cohort basis?

Section 2.8: Discount range should not be varied in sensitivity analysis because there is no intrinsic uncertainty in that parameter.

Results:

It would be easier to read the results table if they were ordered by increasing QALYS instead of decreasing.

Again, framing of scenarios is confusing. Are you comparing drug regimens within each treatment strategy, treatment strategies within each drug regimen, or both?

Based on Table 1, treating acute is cost saving and more effective for DCV+ASV regimen (we usually call this a dominant strategy), but not for PegIFN. For PegIFF it dominates F0, but F3. You can calculate an ICER for F3 vs acute for PegIFF. This is different that what you state in section 3.1.1.

What is the CER in Table 1? If it is incremental, what is the comparison strategy? If it is average, it’s generally not best proactive to report because these ratios can be misleading. ICER for non-dominated scenarios should be shown here.

Section 3.1.2: It would be better to describe with scenarios are dominated. It looks like for DCV+ASV, Both f) and F3 are dominated by actue. For PegINF, F0 is dominated, but F3 and acute are not. Then you only compare non-dominated strategies, so you should calucate an ICER for acute vs F3 for PegIFN, and no ICER for DCV+ASV. If you want to comare between drug regimens, acute DCV+ASV dominates everything else. I would recommend ot calucalting ICERs for dominated strategies.

Figure 2: The lines for GDP are misleading as represented. The DGP line should be compared to the slopes of the comparisons, or the ICER. As represented, where the points are relative to  the DGP lines is meaningless. I would recommend removing these lines from the plot.

Section 3.2.1: It appears sentence 1 is not true, treating at stage f3 with PegIFN had lower costs than acute.

I am unclear of the separation between section 3.2.1 and 3.2.2, what is the difference between these?

Section 3.3: Similar to above, only describe ICERs for non-dominant strategies.

Discussion:

Line 310: It is misleading to say this model included transmission. It included a static effect on reinfection, but no dynamic modeling of transmission. This should be edited in the introduction as well to be more clear.

Reviewer 2 Report

This is a solid piece of evidence filling an important knowledge gap on treatment of Chinese Hepatits C patients with  genotype 1b treatment-naïve patients who injected 24 drugs (PWID), using combination Daclatasvir (DCV) plus Asunaprevir (ASV) regimen and Peg25 interferon (PegIFN)-based regimen.

It has been conducted in an appropriate methodological framework.

Its main weakness is the need to expand evidence base with related evidence coming from LMICs countries and Emerging Markets.

These populations and health sectors are far more similar to Chinese one than Western ones, and should be given due attention.

For this purpose I recommend adoption of at least several of beneath listed sources:

Fitzmaurice, C., Allen, C., Barber, R. M., Barregard, L., Bhutta, Z. A., Brenner, H., ... & Fleming, T. (2017). Global, regional, and national cancer incidence, mortality, years of life lost, years lived with disability, and disability-adjusted life-years for 32 cancer groups, 1990 to 2015: a systematic analysis for the global burden of disease study. JAMA oncology3(4), 524-548.

Su, J., Brook, R. A., Kleinman, N. L., & Corey‐Lisle, P. (2010). The impact of hepatitis C virus infection on work absence, productivity, and healthcare benefit costs. Hepatology52(2), 436-442.

Jakovljevic, M., Jakab, M., Gerdtham, U., McDaid, D., Ogura, S., Varavikova, E., ... & Getzen, T. E. (2019). Comparative financing analysis and political economy of noncommunicable diseases. Journal of medical economics, 1-6 https://doi.org/10.1080/13696998.2019.1600523

Conditional to these minor improvements with or without minor manuscript adjustments I would be willing to embrace revised version for publishing.

Reviewer 3 Report

This is a well-written paper, addressing an area of great importance.  I have a few recommendations that would hopefully improve its message:

1. Whilst the model structure appears to be reasonable, it might be difficult for people without expertise in HCV pathways to feel confident that it captures the right outcomes.  The authors should add a little more detail than simply saying that it was based on previously published papers.  For instance, discuss how the previous models structures look, and how they were received by decision makers and clinicians.

2. It is never appropriate to show 'cost-effectiveness ratios' (as shown in Table 1).  These should be incremental cost-effectiveness ratios.  That is, the difference in costs divided by the difference in benefits from the previously ranked intervention (after dominated and extended dominated interventions have been removed).  Reporting (non-incremental) CERs is incorrect methodology.

3. Based on the comment above, it is not appropriate to say that an intervention is the 'most' cost-effective treatment.  Assuming that all strategies are mutually exclusive, only one intervention should be cost-effective.

4. Likewise, it is not clear in Figures 3 and 4 how the 'most' cost-effective treatment was selected.  See point 2 above - were the studies ranked by effectiveness and then all dominated and extended dominated interventions removed, before applying the decision rule (threshold)?

5. Whilst I am aware that the WHO recommends using GDP and 3x GDP as thresholds, other sources have shown that this is a flawed approach (see: https://www.york.ac.uk/media/che/documents/papers/researchpapers/CHERP109_cost-effectiveness_threshold_LMICs.pdf for more information).  It would be useful for alternative thresholds to be used as well.

6. The discussion section is very useful in highlighting the limitations of the study.  However, it would be even more useful if the authors could go one step further and discuss the implications of those limitations (i.e. might they have changed the conclusions and, if so, in what direction?).
